# Infantile Haemangioma in the Romanian Paediatric Population—Characteristics and Therapeutic Approaches

**DOI:** 10.3390/children10081314

**Published:** 2023-07-30

**Authors:** Anca-Maria Raicu, George-Florin Danila, Ionut Fernando Secheli, Eugenia Claudia Bratu, Dana Galieta Minca

**Affiliations:** 1Department of Public Health and Management, “Carol Davila” University of Medicine and Pharmacy, 050463 Bucharest, Romania; eugenia.bratu@umfcd.ro (E.C.B.); dana.minca@umfcd.ro (D.G.M.); 2Department of Paediatric Surgery, Emergency Hospital for Children “M.S.Curie”, 041451 Bucharest, Romania

**Keywords:** infantile haemangioma, propranolol, timolol, vascular anomalies, paediatric surgery

## Abstract

Infantile haemangioma (IH) is the most common benign tumour in childhood, with an incidence of 4% to 12%. Aim: to describe the characteristics of infantile haemangioma in a sample of Romanian children <2 years old at diagnosis, types of treatment applied, recorded complications and the response to the therapeutic approach. A two-year prospective case series study (August 2019 to August 2021) was carried out. Sample: 117 patients <24 months of age diagnosed with IH at the Emergency Hospital for Children “Marie Sklodowska Curie”, in Bucharest, Romania. Five therapeutic approaches were used: oral treatment with propranolol, local treatment with timolol, surgical treatment, topical treatment with steroids and no treatment (“wait and see”). Recorded factors mentioned in the literature were also present in this study population: female patients—68.4%; phototype I—58%. In 53% of cases, IHs had a head and neck location and 10% developed local complications (traumatic bleeding). The majority of patients (86%) required one type of therapy: oral propranolol (51%). A low relapse rate was recorded (4%). We consider that any child with a vascular anomaly should be referred to a highly specialised medical service for therapeutic approach.

## 1. Introduction

According to the International Society for the Study of Vascular Anomalies (ISSVA), vascular anomalies in infants and children can be classified into two broad categories: vascular tumours and vascular malformations [1].

Haemangiomas belong to the class of vascular tumours, and according to the same Society, can be divided into infantile and congenital [1]

Congenital haemangiomas are rarer, less understood, and are present from birth. They either involute rapidly, in a very short period of time (rapidly involuting congenital haemangioma (RICH)), partially involute (PICH), or never involute (non-involuting congenital haemangioma (NICH)) [2,3,4,5,6].

Infantile haemangioma (IH) is the most common benign tumour in childhood, with an average incidence of 4% to 5% [7,8,9] and a maximum incidence of 10–12%, found in non-Hispanic Caucasian infants. Regarding the gender distribution of patients, it has been observed that the females are most affected [8,10]. Other associated factors for infantile haemangioma described in the literature are: prematurity, low birth weight, multiple pregnancies and skin phototype I [10,11]. At the same time, IH is also considered the most common vascular tumour, which occurs by rapid division of endothelial cells, its growth being attributed to hyperplasia of these cells [11].

In most cases, childhood haemangiomas have a predictable course. Frequently, they are not present at birth (although precursor lesions may be present), but develop in the first weeks of life [12]. Infantile haemangiomas usually appear in the postnatal period and have three evolutionary phases: proliferative, when growth is accelerated, until the age of 1 year, the stationary or plateau phase and the involution or regression phase. The last phase extends until the age of 4–5 years, according to some authors, or 7–12 years, according to others [13]. In most cases, the growth is faster in infants aged one to three months and gradually decreases after the age of five months [14]. Classically, it presents in the form of a mass or stains cutaneous red, of a subcutaneous mass or, generally, of a mixed form associating the two aspects [15,16]. Infantile haemangiomas are usually harmless and can resolve spontaneously and gradually after the age of 1 year, without any intervention [17]. However, about 5–10% of haemangiomas have an unfavourable evolution, with the appearance of telangiectasias, unsightly scars, ulcers and hyperpigmentation [18].

The vast majority of haemangiomas are small lesions, easily recognisable by their clinical features. They most often occur on the skin, but can also affect the viscera (mostly on the liver). Facial haemangiomas can be impressive in appearance, disfiguring, sometimes affecting visual function, causing eating disorders (those of the lips and tongue). IH of the respiratory tract can be life threatening, especially in the proliferative phase. They can appear as single or multiple lesions, they can be simple or part of some syndromes (e.g., PHACE syndrome) [6,19,20]. 

Given the high heterogeneity of haemangiomas, the decision as to who needs treatment, as well as when to start treatment, requires a good knowledge of the natural history and a clinical assessment of the benefit/risk ratio, with a special focus on high-risk predictors. The treatment choice considers the potential evolution, the consequences or prognosis and the impairment generated by the tumour. 

Currently, the treatment of childhood haemangiomas includes a wide range of approaches, from conservative monitoring—“wait and see”—to emergency surgery for severe morbidity and complications. For the haemangiomas showing spontaneous involution, “active non-intervention” is the gold standard and consists of periodic checks, by observing their evolution. To monitor the evolution of the lesion, it is recommended to photograph it at regular time intervals [21].

The absolute indications for treatment are cases with ulceration, disfigurement and impairment of vital organ function [13,14,22,23,24,25,26,27], while relative indications refer to improving the aesthetic appearance, even if the affected body segment is not exposed (e.g., scalp, posterior thorax, gluteal region, etc.).

It is imperative to understand the different phases of growth of IH and what pathological consequences we can expect at each stage, so that treatments can be targeted appropriately and be applied at the optimal time [28,29,30,31].

An important step in the therapy of haemangioma was made in 2008, when the undeniable benefits of propranolol were discovered [32,33,34,35]. Oral propranolol is considered a first-line treatment in patients with proliferating infantile haemangioma at greater than five weeks of age [36,37]. Therefore, when systemic treatment is indicated, propranolol is the drug of choice at a dose of 2 to 3 mg/kg per day initiated between five weeks to five months of age (corrected for premature babies). The treatment is usually continued for at least six months, and it is often maintained until 12 months of age (occasionally longer). Topical timolol may be used to treat small, thin, and superficial IHs [38,39]. Surgery and/or laser treatment are most useful for the treatment of residual skin changes after involution. This is less commonly considered for treating IHs in earlier stages [13,36,40].

## 2. Materials and Methods

The aim of this study was to evaluate the clinical characteristics of infantile haemangioma in the paediatric population under two years of age, the types of treatment applied, and the responses to treatment. 

An observational–prospective, descriptive, epidemiological study design was performed with data collected from patients diagnosed with infantile haemangioma from August 2019 to August 2021, who attended the Emergency Hospital for Children “M S Curie” (Bucharest, Romania). In this two-year period, 117 of the patients diagnosed and treated in the mentioned hospital were included in the study. The selection was made according to an attending physician, yet 90% of the cases were diagnosed, treated and followed up by a single paediatric surgeon. 

Inclusion criteria were: children diagnosed with infantile haemangioma that were younger than 24 months of age, considering that after this age IH enters the regression phase and some treatment options are no longer an option. 

Each patient was examined by the hospital physician, who confirmed the diagnosis of IH. Other forms were excluded accordingly, clinically and on the basis of images, through soft tissue ultrasound examination, transfontanel ultrasound examination for head haemangiomas, abdominal ultrasound examination for abdominal haemangiomas and CT angiography/MRI angiography for selected cases of other vascular abnormalities or dermatological conditions with similar clinical features. 

Personal data (age, gender, area of residence, birth rank, type of birth, gestational age at birth, phenotype), data regarding the overall health status and specific IH information (age at diagnosis, IH localisation and size, treatment types, response to treatment, complications, recurrences) were collected for each child. Data were processed and descriptively analysed with IBM SPSS, version 23.0. 

All subjects gave their informed consent for inclusion before they participated in the study. The study was conducted in accordance with the Declaration of Helsinki, and the protocol was approved by the Ethics Committee of the Emergency Hospital for Children “M S Curie”. (Project identification code—24453/16.07.2019).

## 3. Results

### 3.1. Patients’ Personal Characteristics, Age at Diagnosis and Pregnancy-Related Risk Factors

Of the 117 patients included in the study, 80 were females and 37 were males, with a female: male ratio of 2.2:1. Figure 1 shows the flowchart of the clinical path for this sample. 

A predominance of children with phototype 1 (58%) was observed. Mothers’ mean age at birth was 30 years ± 2.7 years (minimum 15 and maximum 45). 

The distribution of patients according to gestational age at birth is represented in Table 1.

The mean age at the first visit to the hospital was 6.25 months and the median was four months (with the age limits specified in the criteria of inclusion of minimum 0 and maximum 24 months). The age at diagnosis was between 0 and 72 weeks, with a mean of 3.9 (±7.9) weeks and a median of two weeks. Around 79% of children (four out of five) were diagnosed with IH during the first 4 weeks of life; only three cases were diagnosed after more than four months of age. This defines the observed distribution of the age at diagnosis as heavily left-truncated.

Due to the easier access to specialised medical services, most children (69%) came from urban areas. A high proportion (39% or *N* = 46) of patients were from Bucharest.

Although the diagnosis was established early, during the first month of life, only one case immediately presented to the hospital for confirmation of the diagnosis and for targeted, specialised treatment. The mean time difference between the age at diagnosis and the age of presentation to the hospital was 20.6 weeks for the sample, with a median of 16 weeks (*N* = 117). The mean time difference between the age at diagnosis and the age of presentation to the hospital for the cases with complications (*N* = 12) was 28 weeks, with a median of 16 weeks. The mean time difference of 8 weeks at presentation after diagnosis between those who presented with complications (*N* = 12) and those who presented without complications (*N* = 105) was statistically not significant. The CI 95% for the 8 week observed difference in this sample was from 11 to 27 weeks.

In terms of the specialty of those physicians who diagnosed the IH: 53% cases were diagnosed by neonatologists and paediatricians, 20% were diagnosed by the family doctor and 27% by other specialists (dermatologists, paediatric surgeons, otolaryngologists, ophthalmologists, etc). Regarding the pregnancy and maternal factors that could influence the IH occurrence we found out that seven mothers (6%) received hormonal treatments during pregnancy, and none of them were exposed to accidents or gynaecological pathologies during the pregnancy. 

### 3.2. Haemangioma Characteristics 

The number of lesions (isolated tumours) was limited to one or two haemangiomas in 92% of the children (108 out of 117, 95%CI [86.03–95.9]), with most of being them localised on the head and neck (53%) (Table 2). Only in one case was a hepatic haemangioma recorded. Regarding the size of the haemangioma, 45% of the children had a lesion with a maximum size of two cm, and in the case of multiple haemangiomas, the largest of them was accounted for (Table 2). 

Since the occurrence of complications, in the evolution of haemangiomas, represents an absolute indication for a recommended therapeutic scheme, we investigated these thoroughly at the time of presentation. The 12 cases (10%) who presented complications consisted of: ulceration (nine cases) and bleeding (three cases). Complications were the actual reasons for which parents sought medical care and were referred to the paediatric surgeon in the first place. Complications were mostly present in girls. There were nine girls and three boys, therefore in a 3:1 ratio. All 12 cases followed treatment with oral propranolol for up to 12 months, and only one case was treated for longer (18 months), all with good tolerance. They all responded very well to medical treatment and went into remission with minimum sequelae; in one case, laser therapy was recommended (Figure 2a,b).

### 3.3. Therapeutic Options

According to clinical guidance, the available therapeutic options are: oral propranolol, topical β blocker (timolol gel), topical steroids, surgical excision, and “wait and see” (Table 3; Figure 1). Laser therapy, which could not be performed in our hospital, was only recommended for remnant skin lesions, with an onward referral (Figure 2b).

All 117 patients were followed up from time of presentation. Only 14% of the patients had not been treated or were in no need of any treatment at this point (“wait and see” group, *n* = 16, Figure 1). All other patients needed either a single or a combined treatment scheme or plan. Therapeutic management was prescribed in line with each of the patient’s characteristics: the patient’s age, absence/presence of complications, localisation of lesion and its/their dimensions.

A single treatment plan was prescribed for 93 patients: oral propranolol (*n* = 36); topical timolol (*n* = 34), the “wait and see” *(n* = 16), surgical intervention (*n* = 6) and topical steroids (*n* = 1). A fifth of patients (21%) needed to follow a combination of the schemes listed, most frequently a combination of oral propranolol and topical treatment with timolol gel. By specialist indication, the combined treatment schemes were tailored as a combination of either oral propranolol and topical timolol (*n* = 23) or oral propranolol and surgery (*n* = 1) (Figure 1).

In all cases, we recommended an active observation of the IH evolution, through periodic medical reassessments and home monitoring—weekly photos of the lesions, taken by the parent, in the same lighting conditions.

Half (50%) of our patients followed oral treatment with propranolol. The initiation of the treatment was decided in most cases after the consultation and after carrying out the set of specific cardiological and imagistic investigations. The dose was 2 mg/kg/day, divided into two doses. The average duration of treatment was 13 months (minimum 7 months, maximum 24 months), and none of the patients presented with adverse reactions that would have required the cessation of treatment.

In five of the 117 children (4%), a relapse occurred after they stopped the treatment. In all cases, resumption of the treatment was for another 6–7 months, and this led to the total remission of the haemangioma. We noticed that in three of the patients, the signs of relapse appeared during the gradual decrease in the doses, and in the remaining two cases, the relapse occurred four months after the cessation of treatment. Treatment was mainly stopped by clinical indication, always after a Doppler ultrasound examination was used in order to document that there was no vascular signal anymore. There was no recurrent case in which the Doppler ultrasound could, for example, not be performed due to the position of the haemangioma.

None of our patients had adverse reactions that required the interruption of the treatment with propranolol. One patient, who was diagnosed with gastro-oesophageal reflux, resumed treatment after their reflux symptoms improved.

We briefly analysed the parents’ behaviour and the referral pattern to physicians specialised in treating IH. A cultural pattern was observed in parents, especially mothers, whose opinions are driven by superstition in rural areas, e.g., that these ‘injuries’ would appear “as a punishment for the theft of an object or something during pregnancy”. Such superstition leads to the belief that if such a ‘sign’ appears on the skin of the newborn, it will also go away by itself. It is very common that such beliefs prevent parents from seeking medical help for their child in the first place.

## 4. Discussion

Age is an important decision criterion for therapeutic approach. Seventy-two of the diagnosed children (or 62% of the sample) were under five months of age (upper limit of the opportunity window for initiating treatment with propranolol) at the time of the hospital admission. Although prematurity is one of the most cited risk factors for the occurrence of infantile haemangiomas [8,11], only a third of the children were born under a 38-week gestational birth age (34.18%, 95%CI [26.02–43.13]) (Table 1). Having included the gestational age in the analysis, we excluded the weight at birth, as these two variables correlate highly and positively: low gestational age, low birth weight [41]. Four out of five of our patients (79%) were diagnosed in the first month of life, largely by neonatologists or paediatricians, and 62% were under 5 months of age at the time of the examination and diagnosis. Prematurity, considered one of the most important risk factors for IH [8,13,14], was recorded in 34% of our patients (a live birth registered in a mother with a gestational age of less than 38 week). No other factor was linked to the diagnosed IH, e.g., whether the mother had multiple pregnancies (this was <1% in our sample) or whether they had any associated gynaecological pathologies.

Regarding other personal characteristics, we observed a predominance of female patients in our sample (68%), very similar to documented information from the literature; similarly, a slight predominance of children with phototype 1 (58%) was observed [8,9,10,11].

Although propranolol (tablets or oral solution) is considered the gold standard for the treatment of haemangiomas, at the time of the study, up until August 2022, none of the prescriptions were reimbursed by the National Health Insurance House as one would expect, and parents had to cover the treatment as out-of-pocket expenses. Despite the children’s age group, most patients (56%) followed the treatment with propranolol tablets instead of the oral solution, and the main reason for this was financial pressure on the part of the family. Of those parents who could afford the administration of oral propranolol solution from the start of the treatment, only three of them were able to administer it until the completion of the recommended course of treatment. The solution’s price was RON 900 (EUR 185) at the time the study was rolled out, which is much more expensive than tablets were. The oral solution treatment has now been approved by the NHIF under a 6-month cost-volume compensatory scheme in operation since December 2022. Strict criteria are considered: the treatment can be initiated only by specialists from the following categories of medical specialisation: dermatology or dermato-venerology, paediatrics, paediatric cardiology, paediatric surgery, cardiovascular surgery, and haematology. The child’s age must be between 5 weeks and 5 months. Knowing this: IH bears a vital or functional risk, a developmental risk, and a risk of scarring or permanent disfigurement; it is painful, it can ulcerate, bleed or bear a high risk of ulceration and/or bleeding, and it does not respond to simple care of the lesions [42].

Our study has some limitations.

First, the small sample imposed by the COVID 19 pandemic, a period during which the number of patients presenting to the clinic halved when compared with the number of patients who presented with IH in pre-pandemic years. Furthermore, the two-month lockdown (15 March to 15 May 2020) drove the number of presenting patients down even more, according to hospital statistics. However, while this may not have been a further a major limitation, the study design had limitations to begin with, resulting from its agreed and approved research protocol. For example, our design is a descriptive case series of a rare condition. Even if we aimed to design a randomised clinical trial to observe what IH therapeutic line proves more effective for the 0–24 month olds diagnosed with IH, it would have not been possible to design such a study that would be able to detect substantial differences in outcomes (e.g., remission), given the small number of patients, given the low incidence of IH, and given the fixed 2-year study timeline allocation. This is despite the fact that around 79% of children (four out five) were diagnosed with IH during the first four weeks of life, with only three cases reporting a diagnosis after more than four months of age. Thus, by observing the distribution of age at diagnosis being heavily left-truncated, this is a positive feature for IH diagnosis and for access to specialised services. This, however, also gives room for future improvement for the remaining 20% who present later or may not present to a specialist at all. Additionally, the design criteria listed above may never give sufficient power to a multi-arm study aimed at detecting treatment effectiveness as such, or whether a therapeutic line could be defined as superior or inferior when comparative clinical pathways are considered. However, we observed that no adverse reactions or no side effects were recorded in our sample, and that therapeutic plans were well tolerated by all children, including when treatment was interrupted, even just temporarily, in one case.

Secondly, relevant to the incidence and prevalence (when recurrence is registered) of this pathology, only two paediatric surgeons were engaged with the management of all the cases in the study for diagnosis and treatment. We cannot draw conclusions based on the fact that all observations made and recorded during these two years and based on this sample reflect the entire clinical picture of the Romanian IH patients. All new cases of infantile haemangioma are reported by family physicians in Romania, yet reporting is carried out in line with an administrative template, by age group—for under 1 year old (infant) and from 1 to 4 years (child)—and that there are two types of combined pathology i.e., haemangioma and lymphangioma. According to the Romanian National Institute of Public Health statistics, there were 526 new cases of both haemangioma and lymphangioma for children aged 0–4 years in 2019; this means a specific age (0–4 years) incidence rate of 0.051% 95%CI [0.046–0.055]). In relation to diagnosis tools which aid or may speed up the referral process from primary care the new IH Reference Score (IHReS), a two-part algorithm with 12 questions, has been adapted and proposed for use in general practice [43]. This is pending the approval of GP Committees. 

Considering that the 117 patients we treated during the two years may count as one-third of the IH cases referred to our hospital from all over the country, and considering the absence of specific IH clinical guidelines, we draw an important conclusion: that the results of our study are important for the description of the Romanian IH paediatric population including their current clinical pathway and therapeutic management.

Thirdly, we attempted to see whether we could place our findings within contemporary general clinical information available in the literature. Although other studies [17] have shown that, most of the time, no treatment is necessary for IH, our results showed that 86% of our patients needed at least one type of treatment. These results are not surprising, given that patients were referred to our team for specialised IH treatment by physicians of other specialties with limited competences. Similar to other studies [14,15], most patients had haemangioma located on the head and neck (53%), followed by the chest/torso and abdomen (21%) and limbs (20%) (Table 2).

Fourthly, no patterned link was found between the occurrence or presence of complications and the localisation or the size of the haemangioma. Ulcerations are common in areas prone to mechanical friction, e.g., the perianal region, where the areas are susceptible to bleeding, pain, and local infections. The small number of cases that presented with complications of the haemangioma does not allow us to draw firm conclusions in this respect; however, the only observation we made was that all complications were observed in female patients. Complications are usually not life threatening, but have an important emotional impact for the family and for the non-specialised medical personnel. Even if there are effective treatments for complicated haemangiomas, most of the time, they are complex and require a longer treatment time, because the result of an ulceration can become an unsightly scar, which is why we considered it very important that patients diagnosed with infantile haemangioma be directed to specialised services for therapeutic management as soon as possible.

Fifthly, propranolol is the gold standard for the treatment of problematic proliferating IH. Guidelines recommend initiation of propranolol treatment at 1 mg/kg/day, escalating to a maximum empirical dosage of 2–3 mg/kg/day as tolerated, for a minimum of 6–12 months. Response rates of 82–100% to propranolol treatment have been reported with rebound growth of up to 25% and adverse effects of 17–96% with 3% requiring cessation of treatment [32,33,34,35,36]. Clinicians in Romania have aligned their practice with the international recommendation for the treatment of IH since its publication, in 2008. Our results show our professional commitment in lining up with most up-to-date clinical guidelines for the management of IH. That is despite the fact that the administrative component, the health insurance fund, has not stepped up formally to allow a reimbursement scheme and remove the family’s financial burden.

Detailed investigations and the correct diagnosis of IH established before initiating treatment with propranolol are essential for a high success rate. The low recurrence rate (4%) and the lack of severe side effects in children from our selected sample could point to good effectiveness of an already established therapeutic scheme for the therapeutic management in our IH patients. We took note of complaints by pro-actively seeking reactions or effects at the beginning of the treatment for each patient when the dose was still small. There were none, except for two children, and these related to sleep disturbance or agitation, which in the end were proven to be mild single episodes that could not be directly associated with the administration of propranolol, but rather with either age-specific abdominal pain and discomfort or with teething.

We also emphasize the importance of family doctors in the management of cases of IH, hence the recommendation of the use of the IHReS score [43]. We made a note of the 20% proportion (of one in five cases) presented with delay to specialist care after their referral was made by a primary care doctor. Culturally, in Romania, the management of an IH is often dealt with, even by medical professionals, as a condition that “goes away without intervention”. Anecdotally, yet in line with the observed cultural norms on the rural side of residency based on the “wait and see” belief and attitude, such norms can therefore be encountered among medical professionals, too, even when an indication for treatment is very clear. When it comes to peers, primary care physicians, we consider this as a possible consequence of limited access to updated medical information for continuous professional development (CPD). Most other specialties, other than the ones involved in the direct IH specialist care, will have to undergo ‘upskilling’ in what good IH management entails overall in good medical practice. It is important to highlight the fact that in a handful of situations, with one being too many, the therapeutic attitude was not to the benefit of the child. This will be explored in the future after the IHReS has been rolled out. Proof remains in the unexpected development of complications of the IH or, in the delay of the initiation of specialist treatment which can limit the development and aid remission.

Despite these limitations, we consider that our study brings useful information regarding this pathology in the Romanian paediatric population and that it represents a detailed descriptive overview of the current situation of our IH patients in Romania. We draw the conclusion from these limitations that there is a need for efficient data reporting and a reliable outcome monitoring system for the management of this condition. A trend analysis could help with service planning, now that a cost-volume compensatory treatment scheme has lifted the financial burden on families.

## 5. Conclusions

Infantile haemangioma (IH) remains the most common benign vascular tumour. The need for a network of professionals to manage these cases is strong, especially when it comes to the cost of its most effective therapeutic approach. In today’s technological age, parents can easily find on websites or forums incorrect and often worrying information, hence the importance of educating parents of children with haemangiomas to pro-actively seek specialised treatment. Any child diagnosed with a vascular anomaly should be directed to a specialised medical service. The existing information in the literature has proven its importance both for the diagnosis and in the treatment of infantile haemangiomas of children in Romania. Our findings, albeit purely descriptive, highlight the direction for good medical practice of the infantile haemangioma in Romania.

## Figures and Tables

**Figure 1 children-10-01314-f001:**
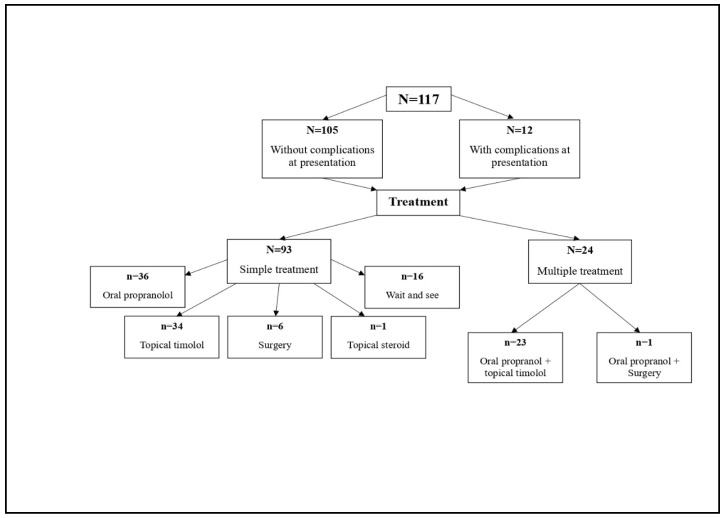
Clinical flowchart of the IH sample (*N* = 117).

**Figure 2 children-10-01314-f002:**
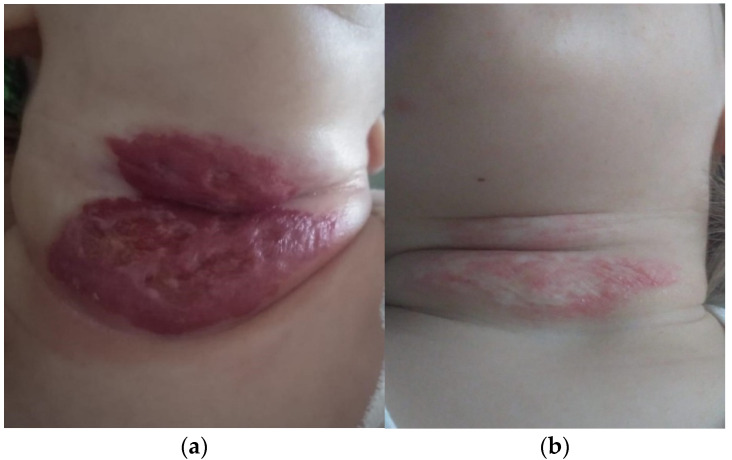
(**a**) Neck IH, 3 months of age. (**b**) After 12 months of treatment with oral propranolol.

**Table 1 children-10-01314-t001:** Distribution of patients according to gestational age at birth (weeks).

Weeks	≤36	37	38	39–41	Total	Mean (Weeks)	Std. Deviation (Weeks)	Median (Weeks)
Frequency N (%)	30 (25.6)	10 (8.5)	32 (27.4)	45 (38.5)	117 (100)	37.33	2.652	38

**Table 2 children-10-01314-t002:** Distribution of patients by lesion type—location, number, dimensions.

Location	Frequency (%)
*Head/Neck*	62 (53.0)
*Torso*	25 (21.4)
*Limbs*	23 (19.7)
*Pampers Area*	7 (6.0)
**Number**	
*1–2*	108 (92.3)
*3–4*	9 (7.7)
**Dimension (cm^2^)**	
*0–2*	53 (45.3)
*2–4*	34 (29.1)
*4–6*	27 (23.1)
*>6*	3 (2.6)
*All*	117

**Table 3 children-10-01314-t003:** Therapeutic approach in children with IH.

	Frequency (%)
**Multiple treatment options**	
*Yes*	24 (20.5)
*No*	93 (79.5)
**Main treatment**	
*Oral Propranolol*	59 (50.4)
*Local Timogel*	34 (29.1)
*Surgical*	7 (6.0)
*No Treatment “Wait and See”*	16 (13.7)
*Topical Corticoid*	1 (0.9)

## Data Availability

The data generated in the present study may be requested from the corresponding author.

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
