# Peer review of "Infantile Haemangioma in the Romanian Paediatric Population—Characteristics and Therapeutic Approaches"

_children, 2023, doi:10.3390/children10081314_

Round 1

Reviewer 1 Report

Comments listed in pdf file

Author Response

Thank you for your assistance with your detailed review responses, queries; we most welcome the improvement suggestions and recommendations to our manuscript.

We now endeavour to have had responded to all queries which were raised, one by one and we resubmit the revised manuscript with the embedded changes and additions as requested. Changes are marked in text with a different colour in the manuscript.

  1. In the “wait and see group”, how many infants developed complications from IH? Did they ultimately require treatment or did their IH resolve on their own? Since the authors are recommending propranolol treatment for all cases, this information is necessary.

The situation was carefully assessed and none of the wait and see developed complications.

Most are still followed up until they reach the age of 24 mo.

The outcome for all 12 cases which presented with complications, given that they were older at presentation, is a clear remission for all of them; mostly were girls.

  1. While propranolol is recommended for treatment of IH, in line 294; Page 7, the authors mentioned about lack of severe side effects on treatment. This statement is vague. What side effects of the treatment was observed in the cases? Since this treatment is used for infants, reporting all observed side effects is important.

In relation to diagnosis tools which aid or may speed up the referral process in primary care the new IH Reference Score (IHReS), a 2-part algorithm with 12 questions, has been adapted and proposed for use in general practice (43). This is pending approval of GP Committees. 

We took note of a couple of complaints of pro-actively sought reactions or effects at the beginning of treatment when the dose was still small. They related to sleep disturbances or agitation which in the end have proven to be mild single episodes which could not be directly associated with the administration of propranolol, but rather with either age-specific abdominal pain and discomfort or with teething.

  1. In discussion, the authors mentioned prematurity as a risk factor for IH, which was only observed in 34% of the patients in this study. Apart from low gestational age, low birth weight is also a known risk factor for IH (ref: Anderson et. al., J Am Acad Dermatol. 2017 ). Did the authors observe this in the cohort? The reference should be added.

We made our discussion point based only on our reference by Hunjan MK, Schoch JJ, Anderson KR, et al. Prenatal Risk Factors for Infantile Hemangioma Development. J Invest Dermatol. 2017;137(4):954-957. doi:10.1016/j.jid.2016.10.047. We now introduce this second reference to mention that low birth weight could still be either a confounding or a collider factor. Our study design does not permit drawing conclusions based on two external studies, but this will be certainly observed in the future and we have included the recommended reference in the Discussion section.

Reviewer 2 Report

Need to improve on the methodology section. Ways on confirmation of diagnosis of infantile hemangioma need to be mentioned. Also the rational of dividing into different treatment modalities need to be mentioned. Techniques involved in different management plans need to be included. Relevant pictures of some sample cases need to be there too. 

The English language is good. 

Author Response

Thank you for your assistance with your detailed review responses, queries; we most welcome the improvement suggestions and recommendations to our manuscript.

We now endeavour to have had responded to all queries which were raised, one by one and we resubmit the revised manuscript with the embedded changes and additions as requested. Changes are marked in text with a different colour in the manuscript.

  1. Is the research design appropriate?

The limitations and practical approach have been described in the Discussion section in detail.

This case series descriptive study design was used as part of a bigger research protocol which mainly looks at access and treatments costs (affordability) for the paediatric population with this rare condition in Romania.

The protocol was approved for a doctorate. The aim of the research, whilst it must at least relate to the exploration of associated or causal risk factors of IH, doesn't include an exhaustive list of confounding or collider variables; we appreciate that while the rigor of an epidemiological study design is important we could not accommodate more variables in this analysis.

The pre-term gestational age did not record a high value in our case-series, therefore we excluded the birth weight from analyses. Birth weight is highly correlated with the gestational age in Romanian live-births. However, this variable will be included in the future in further studies in order to see whether this variable improves the associative or causal information on the development of IH in Romanian live-birth cohorts. We have included the recommended reference in the manuscript (Discussion).

  1. Are the methods adequately described?

The timeline of the study is of two years and also includes patients who were automatically 'discharged' after a follow-up until 24months of age. All the 117 who were enrolled have been so far monitored, new ones have been enrolled but are not included in this study. Monitoring includes the 16 children who were 'prescribed' the wait and see therapeutic management.

  1. Are the results clearly presented?

This section has been entirely revisited.  

We have included Fig 1 the clinical path flowchart. We have included two more figures to show the remission of one case which presented with a complication (Fig 2a and 2b).

We have readdressed content of Tables II and III to match the text better.

We have calculated the mean time difference from diagnosis to presentation for all sample and

We then compared the same time difference (in weeks) calculated for the cases who presented with (N=12) and without complications (N=105).

  1. Are the conclusions supported by the results?

Given that this is a descriptive case series design we consider that our conclusions are supported by these results and we aim to use these findings in relation to the qualitative component of the research protocol (cost of illness, cultural norms, referral system and access to specialist services, quality of life).

Need to improve on the methodology section.

Ways of confirmation of diagnosis of infantile haemangioma need to be mentioned.

Diagnosis was confirmed based on clinical guidance which has been added

Also the rationale of dividing into different treatment modalities need to be mentioned. Techniques involved in different management plans need to be included. Relevant pictures of some sample cases need to be there too.

We have included two anonymised pictures in the resubmitted version of the manuscript.

The rationale of single vs multiple prescriptions included number of lesions, size of lesion(s), whether a complication was or not present and the localization of lesions. The small number of cases does not allow for further development on treatment details at this stage given the study design and even if confidentiality and anonymity are respected such detail would not bring improvement to the results.

We will consider commenting on techniques employed in different management therapeutic plans upon accrual of more future cases. Including additional variables to see whether they make an impact on the treatment results.  

Reviewer 3 Report

Hemangiomas are a significant epidemiological issue. Their wide distribution and variety of presentations result in inconsistent clinical management. Therefore, I believe that an observational study examining the management of these conditions is useful for the literature. The study is well structured and methodologically adequate. The article is well written in all its parts, in particular: the introduction is complete, the methods well explained, the results clear, the conclusions consistent with the results. I appreciated the large space in the discussion for the limitations of the study, I believe they can help in planning higher impact studies in the future. I only have one request for the authors: I think it would be useful to include a representative iconographic part of the presentations of this pathology. Thank you.

Author Response

Thank you for your assistance with your detailed review responses, queries; we most welcome the improvement suggestions and recommendations to our manuscript.

We now endeavour to have had responded to all queries which were raised, one by one and we resubmit the revised manuscript with the embedded changes and additions as requested. Changes are marked in text with a different colour in the manuscript.

  1. Are all the cited references relevant to the research?

We have further added three more references

One recommended reference (Anderson et. al., J Am Acad Dermatol. 2017) and a second to reflect the cost-volume compensatory treatment scheme which was not available during this study, the scheme came into practice after most of our sample came out of the observation period. New patients benefit from the new scheme. The third one related to IReS, useful in the management of this pathology.

  1. Are the results clearly presented?

We agree with the fact that hemangiomas are an important medical diagnosis and that their successful treatment is not only clinically important but the main priority is the quality of life of the patient. New therapeutic guidelines and plans assist in this respect.

Our first case series has had attentive observation and monitoring with the aim to improve access to specialist good quality treatment. We address the importance of financial resources as well as quality of life in a further manuscript.

Round 2

Reviewer 2 Report

Thank you for making the necessary changes. With these changes I suggest for acceptance of the article.